# Protective Mechanism of *Rosa roxburghii* Tratt Fermentation Broth against Ultraviolet-A-Induced Photoaging of Human Embryonic Skin Fibroblasts

**DOI:** 10.3390/antiox13030382

**Published:** 2024-03-21

**Authors:** Minglu Yuan, Hao Fu, Qiuting Mo, Shiwei Wang, Changtao Wang, Dongdong Wang, Jiachan Zhang, Meng Li

**Affiliations:** 1Key Laboratory of Cosmetic, China National Light Industry, College of Chemistry and Materials Engineering, Beijing Technology and Business University, Beijing 100048, China; 2230401038@st.btbu.edu.cn (M.Y.); 1930101019@st.btbu.edu.cn (H.F.); 2130041028@st.btbu.edu.cn (Q.M.); wangct@th.btbu.edu.cn (C.W.); wdd@btbu.edu.cn (D.W.); zhangjiachan@th.btbu.edu.cn (J.Z.); 2Institute of Cosmetic Regulatory Science, Beijing Technology and Business University, Beijing 100048, China; 3Key Laboratory of Resources Biology and Biotechnology in Western China, Ministry of Education, College of Life Science, Northwest University, Xi’an 710069, China; wangsw@nwu.edu.cn

**Keywords:** *Rosa roxburghii* Tratt, *Lactobacillus* fermentation broth, anti-photoaging, human embryonic skin fibroblasts

## Abstract

This study takes the fruit of *Rosa roxburghii* Tratt (RRT) as a fermentation substrate and carries out a quantitative visual analysis of the domestic and foreign literature on screenings of five different lactic acid bacteria to obtain a fermentation broth. Systemic anti-photoaging effects are analyzed at the biochemical, cellular, and molecular biological levels. DPPH and ABTS free radical scavenging activities are used to verify the antioxidant capacity of the RRT fruit fermentation broth in vitro. Human embryonic skin fibroblasts (HESs) are used to establish a UVA damage model, and the antioxidant capacity of the RRT fruit fermentation broth is verified in terms of intracellular reactive oxygen species (ROS) and antioxidant enzyme activity. RT-qPCR and ELISA are used to detect the expression of TGF-β/Smad, MMPs, and the MAPK/AP-1 and Nrf2/Keap-1 signaling pathways in order to explore the anti-oxidation and anti-photoaging effects of the RRT fruit fermentation broth by regulating different signaling pathways. The results show that an RRT fruit fermentation broth can effectively protect cells from oxidative stress caused by UVA and has significant anti-photoaging effects, with the co-cultured *Lactobacillus* Yogurt Starter LYS-20297 having the highest overall effect.

## 1. Introduction

*Rosa roxburghii* Tratt (RRT) is a deciduous shrub in the *Rosaceae* family that is mainly distributed in Guizhou, Sichuan and other places in China. It is a kind of medicinal and edible plant [1,2]. RRT is rich in many active substances such as vitamin C, polysaccharides, flavonoids, polyphenols, organic acids, etc. [3,4,5]. Modern medical research has shown that RRT has a variety of biological effects, such as immune function regulation, sedation, delaying aging, and anti-tumor effects. Among them, its excellent antioxidant capacity has received much attention [6,7,8].

In recent years, probiotic fermentation has become a popular process for fruit product processing [9], as it can increase the content of active substances in fruits, promote the absorption of effective substances, and improve the nutritional value [10,11]. Studies have found that the fermentation of *Passiflora edulis* Sims peel using *Saccharomyces cerevisiae* can more effectively enhance the content of active substances compared with water extraction, and the fermentation liquid shows a superior free radical scavenging ability and antioxidant capacity [12]. Another study used three different probiotics to ferment blueberry juice and blackberry juice, and it found that the metabolism of phenolics increased the antioxidant capacity of blueberries and blackberries and improved their sensory quality [13]. At present, there has been extensive research on the fermentation of RRT, mainly focusing on the changes in the active substances and antioxidant activities in vitro. Some studies have used a variety of different strains to ferment RRT fruit and determine its flavor substances and antioxidant capacities. The test results showed that 87 different flavor compounds were detected and that fermentation also enhanced the free radical scavenging ability of RRT [14].

The skin is the main physiological barrier that protects the body from external damage. Ultraviolet (UV) radiation from the sun is generally considered one of the most dangerous environmental factors for the skin. Current studies have shown that UVA (315–400 nm) is the main factor leading to photoaging [15,16]. The electromagnetic energy from UVA can be absorbed by cell chromophores, and these excited chromophores react with molecular oxygen to produce reactive oxygen species (ROS). Excess ROS causes the oxidation of cellular components, including proteins, lipids, DNA, and RNA, damaging the three-dimensional structures that cells and tissues form; alters the elastic fibers and collagen fibers of the connective tissue of the dermis; and causes abnormal skin structure, relaxation, atrophy, a leather-like appearance, wrinkles, and reduced elasticity [17], ultimately leading to photoaging. Studies have shown that ROS can affect the synthesis of procollagen [16,18]. UVA radiation damages the transforming growth factor beta (TGF-β) molecular pathway, resulting in decreased Type I procollagen [19,20]. ROS can also induce extracellular matrix (ECM) degradation by inducing increases in MMP1, MMP2, MMP3, and MMP9 in the matrix metalloproteinase (MMP) family [21]. The expression of MMPs can be induced by the cascade phosphorylation of proteins. Mitogen-activated protein kinase (MAPK) is a well-known ROS-sensitive signaling pathway, the phosphorylation of which stimulates MMP gene transcription via activator protein-1 (AP-1) [22,23]. The cells can maintain a REDOX steady state by activating enzyme defense systems against ROS, such as superoxide dismutase (SOD), catalase (CAT), and glutathione peroxidase (GSH-px), through signaling pathways such as nuclear factor erythroid 2-related factor 2 (Nrf2)/antioxidant response element (ARE) [24,25].

Thus, clearing ROS and controlling the expression of these signaling pathways may be key to preventing UV-induced photoaging. In this study, lactic acid bacteria were used to prepare the fermentation broth of RRT. The anti-photoaging effects of the fermentation broth of RRT were investigated on the biochemical, cellular, and molecular biological levels, and the expressions of TGF-β/Smad, MMPs, and MAPK/AP-1 and Nrf2/Keap-1 signaling pathways were analyzed so as to explore the anti-photoaging mechanism of RRT fermented by lactic acid bacteria.

## 2. Materials and Methods

### 2.1. Materials

Human embryonic skin fibroblasts (CCC-ESF), CCC-ESF-1, were obtained from the National Collection of Authenticated Cell Cultures; BCA Protein Assay Kit, Lipid Peroxidation MDA, T-AOC, ROS, SOD, CAT, GSH-px Assay kits, Trizol (Total RNA Extraction Reagent)—Shanghai Beyotime Biotechnology (Shanghai, China); DPPH—TCI (Shanghai) Development Co., Ltd. (Shanghai, China); 0.05% (containing EDTA) trypsin, DMEM medium powder, Fetal Bovine Serum, 96-well plate, 6-well plate—Thermo Fisher Scientific (China) Co., Ltd. (Shanghai, China); Viscometers—AMETEK Brookfield (Beijing, China); PH meter,—Eutech Instruments (Shanghai, China); FM medium—Peking Union Medical College Hospital (Beijing, China); ELISA Kit—Beijing Solarbio Science & Technology Co., Ltd. (Beijing, China); ReverTra Ace^®^ qPCR RT Kit—Toyobo (Shanghai) Biotech Co., Ltd. (Shanghai, China); and Fast Super EvaGreen^®^ qPCR Master Mix—Biorigin (Beijing) Inc. (Beijing, China).

### 2.2. RRT Fermentation

RRT dried fruit was purchased from Guizhou Eco-Qianhuo E-commerce Co., Ltd. (Guizhou, China) in Guiyang, Guizhou. It was naturally sun-dried without any other ingredients. The dried RRT fruit was physically crushed and screened through a 50-mesh sieve to obtain RRT powder. A total of 15 g of RRT powder was weighed, and 300 mL of deionized water was added and stirred at 37 °C for 12 h to obtain the supernatant, which was pasteurized (72 °C, 2 h) to obtain the water extract of RRT. Lactic acid bacteria fermentation-related literature [26,27,28,29] was referenced for fermentation strains, and commonly used strains were screened out: *Lactobacillus paracasei*, strain No. CICC-20241, test No. LP-20241; *Lactobacillus helveticus*, strain No. CICC-20243, test No. LH-20243; *Lactobacillus kefiri*, strain No. CICC-20260, test No. LK-20260; Plant *Lactobacillus* species No. CICC-20261, test No. PLS-20261; all were purchased from the China Center of Industrial Culture Collection. *Lactobacillus* Yogurt Starter, strain No. CX-20200907, test No. LYS-20297, provided by Beijing ChuanXiu International Trade Co., Ltd. (Beijing, China), is mixed Lactobacillus, including *Lactobacillus bulgaricus*, *Streptococcus thermophilus*, *Lactobacillus acidophilus*, etc. Activated bacterial broth was obtained by culturing in MRS liquid medium at 37 °C and 180 rpm. When the absorbance at 600 nm was measured to be 1.2, it indicated that the strain had reached the logarithmic growth phase, which was the appropriate fermentation concentration and could be used for inoculation culture. Fermentation was carried out for 24 h at 37 °C with 15 mL of bacteria. After fermentation, the supernatant was obtained and pasteurized. Then, it was subjected to suction and filtration, and finally lyophilized to obtain a lyophilized powder of RRT fermentation broth [30].

### 2.3. Measurement of Total Sugar, Reducing Sugar, Flavonoids, Polyphenols, and Protein Content

The total sugars were detected using the phenol-sulfuric acid method, and the specific operational steps are shown in reference [31]. A total of 1 mL of the test solution and 1 mL of water were mixed, followed by the addition of 0.5 mL of 5% phenol and then 2.5 mL of concentrated sulfuric acid. After thorough shaking, the mixture was allowed to stand for 5 min, and the absorbance was measured at a wavelength of 490 nm in a boiling water bath for 1 h.

The DNS (3,5-Dinitrosalicylic Acid Colorimetry) detection method was used for detecting reducing sugars, and the specific operational steps are shown in reference [32]. A total of 2 mL of the test solution and 1.5 mL of DNS reagent were taken, shaken, heated in a boiling water bath for 5 min, removed immediately, cooled quickly, and then water was added to a constant volume of 25 mL. The mixture was shaken, and the absorbance was measured at a wavelength of 520 nm.

The flavonoid content was determined using the nitrite color development method, and the specific operational steps are shown in reference [33]. A total of 1 mL of the liquid to be tested was taken, 0.3 mL of 5% NaNO_2_ solution was added and shaken, then 0.3 mL of 10% AlNO_3_ solution was added after being left for 6 min. After being shaken, 0.3 mL of 1 mol/L NaOH solution was added, followed by 0.4 mL of water. After being shaken, the solution was left for 10 min. The absorbance of the solution was measured at a wavelength of 510 nm.

The polyphenol content of RRT was measured using the Folin–Ciocalteu method, and the specific operational steps are shown in reference [34]. A total of 1 mL of the solution to be tested was added to 5 mL of 10% Folin-phenol reagent, shaken, and allowed to react for 3–8 min. Then, 4 mL of 7.5% Na_2_CO_3_ solution was added and left at room temperature for 60 min. The absorbance was measured at 765 nm.

A BCA protein assay kit was used for protein content detection, and the specific operation steps are shown in the BCA Protein Assay Kit of Beyotime Biotech Inc. (Shanghai, China). A total of 20 μL of the solution to be tested was added to 200 μL of BCA working solution, which was placed at 37 °C for 15–30 min, and the absorbance was measured at a wavelength of 562 nm.

### 2.4. In Vitro Antioxidant Activity Analysis

For the DPPH free radical scavenging experiment procedures, refer to the method in reference [35]. Experiments were performed using Vitamin C (VC) as a positive control.

For the preparation of the ABTS free radical solution, refer to the Total Antioxidant Capacity Assay Kit with ABTS Method (T-AOC Assay Kit) from Beyotime Biotech Inc. ABTS and oxidants were mixed in a 1:1 ratio and stored overnight at room temperature away from light. On the second day, the mother liquor was diluted 30–50 times. The absorbance of 0.8 at 734 nm could be determined.

### 2.5. Cell Culture

The cell culture bottle was taken out, and the cells were cleaned twice with PBS (pH 7.2). Each bottle of cells was given 0.5 mL of trypsin and placed in the cell culture box for 2 min until the cells were completely digested and suspended. After that, 1 mL of DMEM medium with serum was added to terminate the digestion of the trypsin, and the cell suspension was transferred to a 15 mL centrifuge tube for 5 min at 1500 rpm. After the supernatant was discarded, the cells were precipitated in DMEM medium with serum and blown evenly. They were then transferred to a T25 culture bottle and cultured for two to three days in an incubator at 37 °C with 5% CO_2_; the subsequent experiment could be carried out when the cell confluency rate exceeded 80% [36].

### 2.6. Sample Preparation for Cell Experiments

Preparation of supernatant of cell culture medium: Human fibroblasts in the logarithmic growth phase were counted, inoculated in a 6 cm Petri dish with 2 × 10^6^ cells per well, and cultured overnight at 37 °C and 5% CO_2_. The medium was then discarded, different samples of the same concentration were added to the cells for 24 h, and a serum-free DMEM culture medium was added to the model group and blank control group. After 24 h, the culture medium was removed and a small amount of PBS (pH 7.4) was added to just cover the cells. The cells were exposed to a UVA light source for 5 to 6 min, so that the total dose of irradiation was about 500 mJ/cm^2^, and the control cells were not irradiated. The serum-free DMEM added at the end of the exposure was the supernatant of the cell culture medium.

Preparation of supernatant from cell lysate: IP lysate was prepared at an IP:PMSF ratio of 100:1. After cell culture, the supernatant from the culture medium was collected, and the cells were washed with PBS pre-cooled at 4 °C 2–3 times. The PBS was discarded, and 500 μL of lysis buffer was added to each well. When the cells were fully lysed, the cell lysate and cell precipitates were collected with a cell spatula and centrifuged for 20 min at 12,000 rpm. The supernatant was then collected for later use.

### 2.7. Assay of Cell Viability

A CCK-8 kit was used to measure the cell viability. For the specific experimental steps, see the instructions of the Cell Counting Kit-8 of Beyotime Biotech Inc. A total of 8000 cells (100 μL per well) were seeded in 96-well plates and cultured overnight at 37 °C in a 5% CO_2_ environment. The medium was discarded and different samples with the same concentration were added to culture for 24 h. Then, 10 μL CCK-8 reagent was added to each well and incubated for 2 h at 37 °C, and the absorbance was measured at the wavelength of 450 nm.

### 2.8. ROS Content Detection

ROS content was determined using the ROS Assay Kit of Beyotime Biotech Inc. For the specific operation steps, see the manufacturer’s manual. The positive control was 50 μg/mL of Vitamin C (VC), the cells were digested with trypsin after the addition of the probe, and the fluorescence values were detected by a fluorescent microplate reader or flow cytometry.

### 2.9. Enzyme Activity Assay

CAT, GSH-px, and SOD enzyme activities were detected using a Catalase Assay Kit, Cellular Glutathione Peroxidase Assay Kit with NADPH, and Total Superoxide Dismutase Assay Kit with NBT of Beyotime Biotech Inc. For the specific experimental steps, please refer to the manufacturer’s instruction manuals. The CAT and GSH-px assay kits were used to detect the supernatant of the cell lysate, while the SOD assay kit was used to detect the plasma lysate cells of the sample provided by the kit.

### 2.10. Total Antioxidant Capacity and Lipid Peroxidation Level

The total antioxidant capacity and lipid oxidation level of the cells were determined using the Total Antioxidant Capacity Assay Kit with ABTS Method and Lipid Peroxidation MDA Assay Kit of Beyotime Biotech Inc. For the specific experimental procedures, see the manufacturer’s instruction manuals. All samples tested by the kit were homogenized lysate cells provided by the kit.

Total Antioxidant Capacity Assay Kit with ABTS method: In the 96-well plate, 200 μL ABTS working solution and 10 μL test solution were added, gently mixed, and incubated at room temperature for 2–6 min before measuring the absorbance value at 734 nm.

Lipid Peroxidation MDA Assay Kit: 0.1 mL of the test solution was added to 0.2 mL of the MDA detection working solution, which was mixed and then bathed in boiling water for 15 min, cooled to room temperature, centrifuged at 1000 g for 10 min, and the absorbance was measured at a wavelength of 532 nm.

### 2.11. Enzyme-Linked Immunosorbent Assay (ELISA)

Intracellular protein (MMP-1, COL-I, ELN) was detected using a Human ELISA Kit of Beijing Solarbio Science & Technology Co., Ltd. (Beijing, China). For the specific experimental steps, see the manufacturer’s instruction manual. The test samples of the kit were supernatant cell culture medium.

### 2.12. Real-Time Quantitative Polymerase Chain Reaction (RT-qPCR)

Total RNA extraction was carried out based on the Trizol reagent of Beyotime Biotech Inc. For the specific extraction method, see the manufacturer’s instructions. The first strand of cDNA was synthesized using a ReverTra Ace^®^ qPCR RT Kit of Toyobo (Shanghai) Biotech Co., Ltd. (Shanghai, China). See the manufacturer’s instruction manual for the specific reaction steps.

According to the gene sequences published in NCBI, the primers using β-actin as the internal reference gene were designed. The specific primer sequences of a total of 15 genes are shown in Appendix A.

After the total RNA was extracted, PCR was performed according to a Fast Super EvaGreen^®^ qPCR Master Mix of Biorigin (Beijing) Inc. (Beijing, China). The total reaction system was 20 μL. Refer to the manufacturer’s instruction manual for the specific reagent dosage.

Cycle parameters: Pre-denaturation was conducted at 94 °C for 30 s, 94 °C for 15 s, 60 °C for 15 s, and 72 °C for 10 s, for a total of 40 cycles, to collect the fluorescence data. The reactions were performed on a QuantStudio™ 3 Real-Time PCR System of Thermo Fisher Scientific Inc. (Shanghai, China).

### 2.13. RNA Sequencing (RNA-seq)

Total RNA extraction was based on the Trizol reagent of Beyotime Biotech Inc. (Shanghai, China). For the specific extraction method, see the manufacturer’s instructions.

The extracted RNA was sent to Shanghai Majorbio Bio-pharm Technology Co., Ltd. (Shanghai, China). for transcriptome sequencing analysis. The transcriptome data of the cell protection groups were divided into the control group, model group, and RRT fermentation broth group (RFP).

The differential gene expression analysis of the sample genes was performed, as well as GO and KEGG gene function annotation analysis and gene set enrichment analysis. For the specific analysis operations, see the Majorbio Bio-pharm platform.

### 2.14. Data Analysis

All measurements were performed in at least 3 separate experiments, and all values are expressed as mean ± standard deviation (SD). Statistical calculations were performed using the IBM SPSS Statistics 22 software program (Chicago, IL, USA). *p* < 0.05 was considered a statistically significant difference.

## 3. Results

### 3.1. Active Substance Content

The physicochemical properties of the supernatant of RRT fruit fermented by different strains are similar, and the specific values are shown in Table 1. The fermentation supernatant was yellow and clear without precipitation. The whole fermentation broth was acidic, and the pH value of the fermentation broth of different strains was between 4 and 5. The viscosity was all about 17 mPa·S, and only PLS-20261 had a slightly higher viscosity of 19 mPa·S.

First, in vitro biochemical experiments were conducted to detect the active substances in RRT fruit fermentation broth from different strains of bacteria, including total sugars, reducing sugars, flavonoids, polyphenols, and proteins. The experimental results are shown in Table 2. After fermentation by different strains, higher contents of active substances could be obtained in the fermentation broth, and the contents of total sugars, reducing sugars, and proteins could be improved by selecting the lactic acid bacteria yogurt fermentation powder co-cultured by different strains.

### 3.2. Antioxidant Efficacy In Vitro

The antioxidant effects of RRT fruit fermentation broth from different strains of bacteria were compared in vitro, and DPPH and ABTS free radical scavenging activities were specifically detected. In the experiment, DPPH radical scavenging activity of 0.015–1 mg/mL was detected, and a 50% inhibitory concentration (IC_50_) value of the scavenging rate of both was calculated. The experimental results are shown in Figure 1A. The IC_50_ values are as follows: LP-20241 0.151 mg/mL, LH-20243 0.148 mg/mL, LK-20260 0.166 mg/mL, PLS-20261 0.197 mg/mL, and LYS-20297 0.169 mg/mL. As can be seen from the concentration curve, there is no significant difference between RRT fruit fermentation broth from different strains of bacteria, all of which have strong free radical scavenging activities.

The total antioxidant capacity of the fermentation broth at the same concentration was compared with that of the total antioxidant kit, and the comparison results are shown in Figure 1B. The results show that the fermented RRT fruit had a high total antioxidant capacity.

### 3.3. Cell Viability and UVA Model Establishment

The CCK-8 experiment was carried out on the RRT fruit fermentation broth from different strains of bacteria, and the experimental results are shown in Figure 2A. The IC_80_ of each sample was 0.563 mg/mL for LP-20241, 0.291 mg/mL for LH-20243, 0.381 mg/mL for LK-20260, 0.205 mg/mL for PLS-20261, and 0.397 mg/mL for LYS-20297 by SPSS calculation. It can be seen that when PLS-20261 was added at 0.205 mg/mL, the cell tolerance was the worst, and a dose less than this critical value was considered safe. Therefore, the IC_80_ concentration of 200 μg/mL was selected as the experimental cell concentration. The samples were added to the detection hole after 24 h, and it was found that the survival rate of the cells had no effect, and even had an enhancing effect. It can be concluded that this mass fraction of the sample has no toxicity to cells and will not affect their growth.

The intracellular ROS content was detected using probe DCFH-DA, and the specific experimental results are shown in Figure 2B. It can be intuitively seen from the microscope photos that the intracellular ROS content was significantly increased after UVA irradiation, while it was significantly decreased under the protection of the sample. To further verify the correctness of the experimental results, the cells were digested and collected with trypsin, then configured as a cell suspension and uniformly added into 96-well plates to measure the fluorescence intensity. The experimental results are shown in Figure 2C. After UVA irradiation, the intracellular ROS content was increased by 160% compared with the blank group. After being treated with RRT fruit fermentation broth from different strains of bacteria, the ROS content was decreased, with LYS-20297 having the most significant decrease.

### 3.4. Intracellular Antioxidant Levels

The intracellular antioxidant levels were studied experimentally, and the enzyme activities of CAT, GSH-px, and SOD were detected, respectively. The experimental results are shown in Figure 3A–C. As can be seen, the activities of the three enzymes in the cells were significantly decreased after UVA irradiation, then increased to varying degrees after the action of RRT fruit fermentation broth from different strains of bacteria. In conclusion, RRT fruit fermentation broth significantly improved the activity of intracellular antioxidant enzymes and effectively protected fibroblasts from UVA-induced oxidative stress.

The total antioxidant capacity of the cells was tested, with ABTS free radical scavenging taken as the standard and Trolox used as the experiment standard. The experimental results are shown in Figure 3D. As can be seen, after UVA irradiation, the total antioxidant capacity of the cells significantly decreased, then significantly increased after sample treatment. In addition, it can be seen that there is little difference in antioxidant capacity across RRT fruit fermentation broth from different strains of bacteria, which is similar to the antioxidant capacity in vitro results mentioned above.

Figure 3E shows the effects of the samples on the lipid oxidation levels of cells under oxidative stress. As can be seen, the intracellular lipid oxidation level was significantly increased after stimulation by hydrogen oxide, while the intracellular malondialdehyde content was significantly decreased after treatment with the RRT fruit fermentation broth from different strains of bacteria. The results show that the fermentation broth could significantly reduce the content of intracellular malondialdehyde and effectively prevent the increase in the intracellular lipid oxidation level.

Finally, based on the cell experiment data, the related efficacy of RRT fruit fermentation broth from different strains of bacteria was sorted. Among them, the highest efficacy was 5 and the lowest was 1, and data heat map F was created. It was found in the heat map that the overall efficacy was relatively the highest for the LYS-20297 fermentation broth. Therefore, it was selected as the research object for the later biomolecular experiment.

### 3.5. Detection of Photoaging-Related Protein Content

The most intuitive manifestation of photoaging is the change in elastic fibers and collagen fibers in the connective tissue of the dermis. Therefore, this paper further tested the influence of RRT fruit fermentation broth on the intracellular collagen content. The intracellular protein levels were detected using an enzyme-linked immunosorbent assay (ELISA) kit. Figure 4A shows the results of Type I collagen (COL-I) content. As can be seen, the COL-I content was significantly decreased after UVA irradiation, from 131 pg/mg to 53.47 pg/mg, while the intracellular COL-I content was significantly increased after treatment with RRT fruit fermentation broth. After treatment with LK-20260, PLS-20261, and LYS-20297, the COL-I content increased to 282, 327, and 414 pg/mg, respectively.

MMPs are mainly used to degrade collagen in the ECM. MMP-1 is a key member of the MMP family, which can initiate collagen decomposition and plays a key role in skin aging. Figure 4B shows the results of the detection of intracellular MMP-1 content. It can be seen that the content of MMP-1 protein was significantly increased after UVA irradiation, while it was significantly decreased after protection by RRT fruit fermentation broth.

Elastin (ELN) and collagen form the main structure of the ECM. ELN is a fibrin that guarantees the elasticity and strength of human skin and is also involved in tissue repair. Figure 4C shows the detection results of intracellular ELN content. It can be seen that the ELN content decreased significantly after UVA irradiation, from 489.67 pg/mg to 289.67 pg/mg. The content of intracellular ELN was then significantly increased after treatment with RRT fruit fermentation broth, with LYS-20297 showing the most significant increase (1253 pg/mg).

### 3.6. The Expression Levels of Collagen Synthesis- (TGF-β) and Degradation (MMP)-Related Genes Were Detected

The TGF-β/Smad signaling pathway regulates the biosynthesis of dermal fibroblasts. When the receptor complex is activated, TGF-β signaling is triggered, which positively regulates the activity of Smad2 and Smad3. RT-qPCR was used to detect mRNA expression levels. Figure 5A shows the results of the relative expression level of the TGF-β gene in cells. As shown in Figure 5A, TGF-β was significantly down-regulated after UVA irradiation (*p* < 0.001), and its relative expression was significantly up-regulated after treatment with RRT fruit fermentation broth (*p* < 0.001). Smad3 is the downstream gene of the TGF-β/Smad signaling pathway, and TGF-β positively regulates Smad. This positive regulation involves the phosphorylation of Smad2 and Smad3, which induces their binding and formation of the Smad2/3 complex. This complex is transferred to the nucleus, where it regulates the transcription of multiple target genes, including Type I procollagen. Figure 5B shows the expression of the Smad3 gene in cells. It can be seen that the Smad3 gene was significantly down-regulated after UVA irradiation (*p* < 0.001), while its expression was significantly up-regulated after treatment with RRT fruit fermentation broth. The expression of Smad7 is also induced by TGF-β, which negatively regulates TGF-β signaling by forming a heteropolymer complex with Smad2/3, thereby inhibiting complex formation. As shown in Figure 5C, the expression of Smad7 was significantly up-regulated after UVA irradiation (*p* < 0.001), while it was significantly down-regulated after treatment with RRT fruit fermentation broth (*p* < 0.001).

Collagen is the main component of the ECM and dermal collagen, including COL-I and Type III collagen (COL-III). A decrease in COL-I and COL-III and an increase in the ratio of COL-I/COL-III are indicators of skin photoaging. Figure 5D,E show the intracellular gene expression levels of COL-I and COL-III. As can be seen, the gene expressions of COL-I and COL-III were significantly down-regulated after UVA irradiation (*p* < 0.001), while they were significantly up-regulated after treatment with RRT fruit fermentation broth.

The main function of MMPs is to degrade collagen in the ECM. MMP-1 mainly plays a role in degrading collagen, while MMP-2, MMP-3, and MMP-9 mainly degrade different collagen fragments. The TIMP gene can inhibit the expression of metalloproteinase genes and alleviate the degradation of collagen in the ECM by MMPs. Figure 5F shows the detection results of intracellular TIMP. It can be seen that the expression of the intracellular TIMP gene was significantly down-regulated after UVA irradiation (*p* < 0.01), while it was significantly up-regulated after treatment with RRT fruit fermentation broth (*p* < 0.001). Therefore, RRT fruit fermentation broth can inhibit the expression of MMP genes.

Figure 5G–J show the relative expressions of the MMP-1, MMP-2, MMP-3, and MMP-9 genes in the cells. On the whole, after UVA irradiation, the expression of the MMP genes was significantly increased (*p* < 0.001), while it was significantly decreased after treatment with RRT fruit fermentation broth (*p* < 0.001). Therefore, it can be concluded that RRT fruit fermentation broth can significantly reduce the expression of intracellular MMP genes to effectively protect the collagen content in cells from degradation.

### 3.7. Transcriptome Sequence Analyses

The protective effects of RRT fruit fermentation broth on UVA-induced oxidative stress damage in fibroblasts were systematically analyzed using the RNA-Seq technique. First, three groups of samples were prepared, namely the blank group, model group, and sample group. The differential genes in the different groups can be seen in the summary of differential genes in Figure 6A. In Figure 6C, target gene cluster analysis was performed on the differential gene sets, and the results show that RRT fruit fermentation broth could significantly increase the expression of intracellular genes when protecting cells from oxidative stress caused by UVA radiation. There were 1933 differentially expressed genes in total, and the specific number of differentially expressed genes is shown in Appendix A. Differentially expressed gene sets were established for target gene annotation and enrichment analysis.

GO and KEGG were used to analyze the differential gene sets in a centralized way. A total of 53 s-level classification terms, totaling 2755 genes, were obtained through GO analysis. Figure 6D was obtained after summarizing the top 20 terms of gene abundance. Among them, the BP group had the largest number of genes involved in cellular processes, with 1639 genes involved. The CC group had the largest number of genes involved in cells, with 1925 genes. The MF group had the largest number of genes involved in binding response, with 1720 genes involved. A total of 319 metabolic pathways were obtained by KEGG analysis, among which the top 20 were screened for abundance according to the number of genes, and finally, Figure 6E was obtained. For specific information on the 20 metabolic pathways, see Appendix A. In the cellular process, 128 genes were involved in cell growth and death; in environmental information processing, 241 genes were involved in signal transduction; in genetic information processing, 64 genes were involved in replication and repair; and in metabolism, 43 genes were involved in carbohydrate metabolism.

The enrichment analysis of their biological functions was conducted to obtain the related pathways, and the enrichment analysis of the GO and KEGG pathways was carried out. Among them, 3061 GO term genes with a total of 2416 genes were found by GO enrichment analysis, as shown in Figure 6F, of which the *p*-values were all <0.05. The focus was on the discovery of 20 genes. The detected genes were mainly involved in the regulation of the binding and action processes between cells, organisms, and molecules, including protein binding and metabolism, ion binding and metabolism, and so on. A total of 323 pathways were enriched by KEGG enrichment analysis, as shown in Figure 6G. Among them, 73 pathways with *p* < 0.05 were detected, and 13 signaling pathways were screened out as related to cell aging. See Appendix A. In addition, previous studies have found that when photoaging occurs, cells will first produce an oxidative stress response, with ROS being its product, and a series of pathways are activated in the cells to remove it. Finally, the genes in Table 3 and their path-related genes were selected for further verification, as well as the up-regulation and down-regulation results of transcription component precipitation genes.

### 3.8. The Expression Levels of Oxidative Stress- (MAPK/AP-1) and Antioxidant Enzyme (Nrf2/Keap-1)-Related Genes Were Detected

According to the transcriptome analysis results, select ROS signaling pathways were removed in cells. The MAPK/AP-1 signaling pathway is commonly used in cellular oxidative stress studies including photoaging, cell proliferation and apoptosis, and inflammation. First, we verified that the constituent genes of MAPK, including JNK, ERK, and P38, were affected by RRT fruit fermentation broth, and the experimental results are shown in Figure 7A–C. As can be seen, the intracellular gene expression was significantly up-regulated after UVA irradiation, while it was significantly down-regulated after treatment with RRT fruit fermentation broth.

AP-1 is an important intracellular transcription factor composed of FOS and JUN which plays a key role in regulating the expression of MMPs. As shown in Figure 7D,E, the expression of AP-1 was significantly up-regulated after UVA irradiation (*p* < 0.001), while it was down-regulated after treatment with RRT fruit fermentation broth (*p* < 0.001).

The Nrf2/Keap-1 pathway is a key pathway that protects cells from oxidative stress and regulates the activity of antioxidant enzymes in cells. According to Figure 7F, the relative expression level of Nrf2 was significantly decreased after UVA irradiation (*p* < 0.01), while it was significantly increased after treatment with RRT fruit fermentation broth (*p* < 0.001). Figure 7G shows the Keap-1 detection results. It can be seen that the Keap-1 gene was significantly up-regulated after UVA irradiation (*p* < 0.001), while it was significantly down-regulated after treatment with RRT fruit fermentation broth.

HO1, CAT, SOD, and GSH-px are all downstream genes of the Nrf2/Keap-1 signaling pathway which are important mechanisms for ROS clearance to protect cells from oxidative damage. As can be seen from Figure 7H–K, the HO1, CAT, SOD, and GSH-px genes were significantly down-regulated after UVA irradiation, while they were significantly up-regulated after treatment with RRT fruit fermentation broth, which plays a role in protecting cells from oxidative stress.

## 4. Discussion

This study took RRT fruit as the fermentation substrate and carried out a quantitative visual analysis of domestic and foreign literature on screenings of five different lactic acid bacteria, namely *L. paracasei*, *L. helveticus*, *L. kefiri*, Plant *L.* species, and *Lactobacillus* Yogurt Starter, to obtain the fermentation broth. The systemic anti-photoaging effects were analyzed at the biochemical, cellular, and molecular biological levels. The results show that RRT fruit fermentation broth can effectively protect cells from oxidative stress caused by UVA, and has significant anti-photoaging effects, with the co-culture mixed LYS-20297 having the highest overall effect. In the past, laboratories usually used isolated and purified single bacteria for fermentation-related experiments, so four different purified lactic acid bacteria were selected as the experimental subjects in this study. However, the genome sequence results of some microorganisms in the past have shown that microorganisms have the potential to produce more active substances than observed in traditional single-species cultures [37]. Thus, many biosynthetic gene clusters may be silent when a single strain is cultured for fermentation [38]. This phenomenon is not surprising, because microbial activities in the natural environment are often subject to complex ecological relationships. The interactions among microorganisms and between microorganisms and the environment form a relatively stable biological community structure that continues to evolve in time and space. Some biochemical processes require more than two kinds of microorganisms to be carried out, and some substances need to be co-cultured by a variety of microorganisms to be produced [39]. Complicating the fermentation environment in the laboratory setting by introducing different strains for co-culture is a useful method that can activate biosynthetic silenced genes and induce the production of novel metabolites [40]. In the present study, the co-culture of Lactobacillus LYS-20297 was used for comparative experiments, and it was confirmed that the fermentation broth of Lactobacillus LYS-20297 contained higher amounts of active substances and had a stronger anti-photoaging ability than other single-strain bacteria.

Studies have shown that excessive ROS can directly lead to cell damage, such as mitochondrial DNA damage, singlet oxygen oxidation of guanine, and protein carbonylation, which can cause cellular oxidative stress and lead to skin aging [41]. Therefore, the elimination of excess ROS and the reduction in cellular oxidative stress are considered powerful means of anti-aging [42]. In vitro biochemical experiments showed that RRT fruit fermentation broth contained abundant active substances such as total sugars, reducing sugars, flavonoids, polyphenols, and proteins. DPPH and ABTS free radical scavenging experiments showed that RRT fruit fermentation broth from different strains of bacteria had strong free radical scavenging effects, and the difference between strains was not obvious.

A UVA-induced fibroblast damage model was then established to study the protective effects of RRT fruit fermentation broth on UVA-induced oxidative stress [43]. In this study, the ROS level in human embryonic skin (HES) cells after UVA irradiation was first detected, and then the protective ability of RRT fruit fermentation broth on fibroblasts was evaluated by detecting the enzyme activity of intracellular antioxidant enzymes such as CAT, GSH-px, and SOD, as well as the total antioxidant capacity and lipid oxidation level in cells. The results showed that RRT fruit fermentation broth could alleviate oxidative damage caused by UVA irradiation, and had strong protective effects.

The most obvious expression of photoaging is the changes in elastic fibers and collagen fibers in the dermal connective tissue [44]. The expressions of MMP-1, COL-I, and ELN were detected by ELISA. It was found that RRT fruit fermentation broth significantly inhibited the synthesis of MMP-1 proteins and increased the contents of COL-I and ELN, thereby protecting the elastic fibers and collagen fibers of the derma. The RT-QPCR results also verified that RRT fruit fermentation broth could improve collagen synthesis in vivo by activating the TGF-β/Smad pathway and inhibiting the expression of MMPs to reduce collagen decomposition, increase the synthesis of dermal collagen, and play a role in photoaging protection [45]. Transcriptomic analysis was then used to obtain clues of anti-photoaging at the molecular level. The activation of the MAPK/AP-1 pathway reduces the synthesis of MMPs in vivo, which lessens the damage they cause to the ECM; the activation of the Nrf2/Keap-1 pathway improves antioxidant capacity in vivo [46]. RT-QPCR confirmed that RRT fruit fermentation broth could achieve antioxidant and anti-photoaging effects by regulating different signaling pathways.

In this study, the cell model used for the experiments was a fibroblast derived from human embryonic tissue. Because cells derived from embryos have the greatest cell viability and proliferation potential, they are suitable for laboratory applications [47]. However, it is well-known that adult skin is more likely to be exposed to sunlight, so possible differences between embryonic and adult cells would be a limitation of this study. On the other hand, this study focused on gene expression changes at the transcriptional level, but the response of cells to changes in environmental conditions is a dynamic process. For example, several signaling pathways have been shown to operate at the level of post-translational modification, which precedes gene expression changes. However, the change in protein levels has a relative lag, and the accumulation of some proteins even takes considerable time. Therefore, more temporal differences should be included in the discussion of studies in the future to observe more realistic cellular activities.

## 5. Conclusions

This study provides evidence that RRT fruit fermentation solution protects HES against photoaging induced by UVA radiation. At the same time, it was proven that the fermentation broth obtained by co-culture mixing Lactobacillus LYS-20297 had a stronger effect overall. Our research has proven that LYS-20297 contains a higher content of active substances than other fermentation liquids, significantly increasing the activity of antioxidant enzymes in cells, reducing the overall oxidative stress level of cells through various signaling pathways, increasing the content of collagen and elastin, and protecting cells to slow down photoaging caused by UVA radiation. However, the specific active substances that play a role in the fermentation solution still need further research.

## Figures and Tables

**Figure 1 antioxidants-13-00382-f001:**
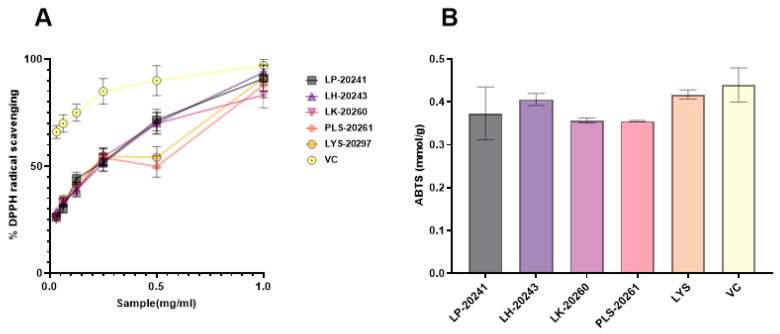
Antioxidant activities in vitro of RRT fruit fermentation broth from different strains of bacteria: (**A**) DPPH free radical scavenging activity; (**B**) ABTS free radical scavenging activities.

**Figure 2 antioxidants-13-00382-f002:**
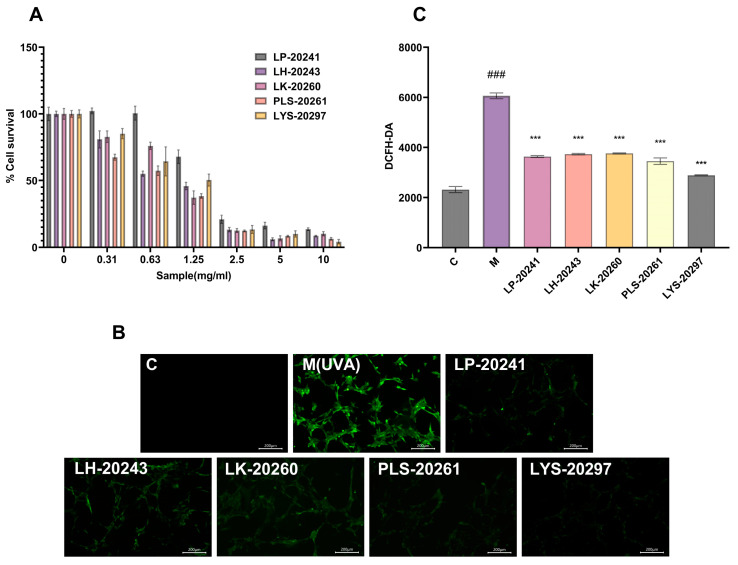
(**A**) Effects of samples on cell viability; (**B**) fluorescence microscope photos (10×); (**C**) fluorescence intensity was detected by fluorescent microplate reader. (###: *p* < 0.001 model group vs. blank group; ***: *p* < 0.001 sample group vs. model group).

**Figure 3 antioxidants-13-00382-f003:**
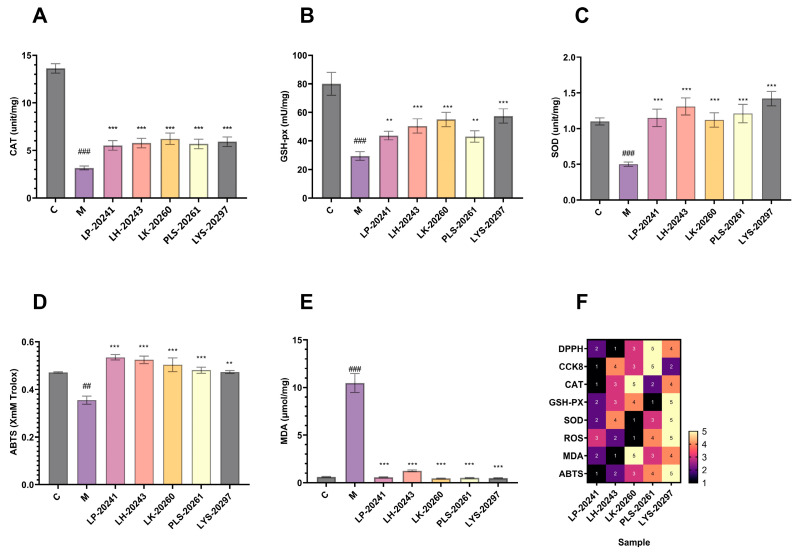
Intracellular (**A**) CAT; (**B**) GSH-px; (**C**) changes in the activity of SOD; (**D**) changes in intracellular total antioxidant capacity; (**E**) changes in intracellular lipid oxidation levels; (**F**) heat map. (##: *p* < 0.01, ###: *p* < 0.001 model group vs. blank group; **: *p* < 0.01, ***: *p* < 0.001 sample group vs. model group).

**Figure 4 antioxidants-13-00382-f004:**
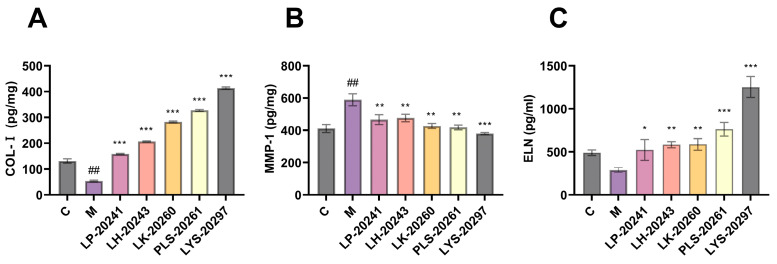
Intracellular (**A**) COL-I protein; (**B**) MMP-1 protein; (**C**) detection results of ELN protein content. (##: *p* < 0.01 model group vs. blank group; *: *p* < 0.05, **: *p* < 0.01, ***: *p* < 0.001 sample group vs. model group).

**Figure 5 antioxidants-13-00382-f005:**
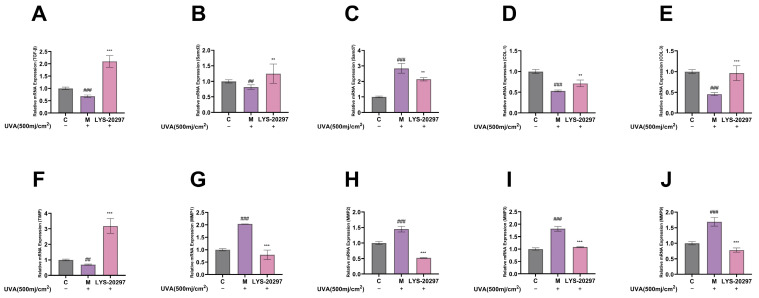
Intracellular gene relative expression levels: influence of RRT fruit fermentation broth on relative expression levels of (**A**) TGF-β gene; (**B**) Smad3 gene; (**C**) Smad7 gene; (**D**) COL-I gene; (**E**) COL-III gene; (**F**) TIMP gene; (**G**) MMP-1 gene; (**H**) MMP-2 gene; (**I**) MMP-3 gene; (**J**) MMP-9 gene. (##: *p* < 0.01, ###: *p* < 0.001 model group vs. blank group; **: *p* < 0.01, ***: *p* < 0.001 sample group vs. model group).

**Figure 6 antioxidants-13-00382-f006:**
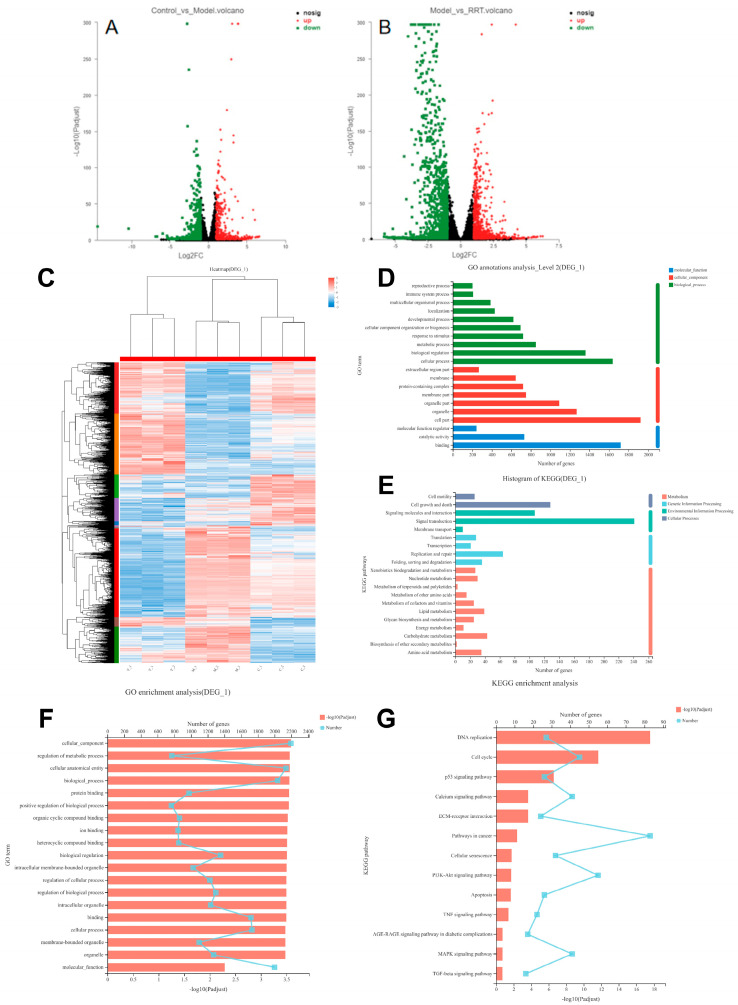
Differential gene volcano map: (**A**) blank group vs. model group; (**B**) model group vs. sample group, in which the horizontal coordinate is the multiple change value of the gene expression difference between the two samples, and the vertical coordinate is the *p* adjust value; the higher the *p*-value, the more significant the expression difference. Dots in the figure represent genes, with red indicating significantly up-regulated genes, green indicating significantly down-regulated genes, and black indicating no significant expression. (**C**) Cluster analysis of differential genes: each column represents a sample, each row represents a gene, and the color in the figure represents the gene expression level in the sample. By default, red represents a higher expression level of the gene or transcript in the sample, while blue represents a lower expression level. For details on the variation trend of the expression level, please see the digital annotation under the color bar in the upper right. (**D**) GO annotation analysis results: the horizontal coordinate represents the specific name of the GO classification, the vertical coordinate represents the number of genes in the GO classification, and the three colors represent the three classifications, namely molecular function (MF), cell composition (CC), and biological process (BP); (**E**) KEGG annotation analysis results: the x-coordinate represents the number of genes in the metabolic pathway, while the y-coordinate represents the name of the metabolic pathway; (**F**) GO enrichment analysis results: the y-coordinate represents the GO term, and the upper x-coordinate represents the number of genes in the GO term corresponding to the broken line point, while the lower horizontal coordinate represents the enrichment significance corresponding to the column height; the higher the *p*-value, the more significant the GO enrichment; (**G**) KEGG enrichment analysis results: the ordinate is the pathway name, the upper abscess is the number of genes in the pathway represented by the broken line chart, and the lower abscess is the enrichment significance represented by the column height; the higher the *p*-value, the more significant the KEGG enrichment.

**Figure 7 antioxidants-13-00382-f007:**
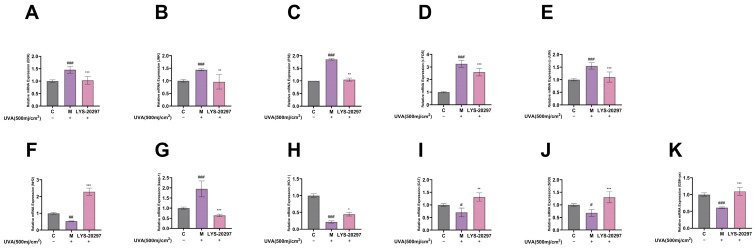
Relative expression of MAPK pathway genes, (**A**) ERK gene, (**B**) JNK gene, (**C**) P38 gene. Intracellular relative expression of AP-1 gene, (**D**) c-FOS gene, (**E**) c-JUN gene. Relative expression levels of genes in cells: (**F**) Nrf2 gene; (**G**) Keap-1 gene; (**H**) HO-1 gene; (**I**) CAT gene; (**J**) SOD gene; (**K**) GSH-px gene. (#: *p* < 0.05, ##: *p* < 0.01, ###: *p* < 0.001 model group vs. blank group; *: *p* < 0.05, **: *p* < 0.01, ***: *p* < 0.001 sample group vs. model group).

**Table 1 antioxidants-13-00382-t001:** Physicochemical properties of fermentation broth of RRT fruit from different strains of bacteria.

Sample	pH	Conductivities (μS/cm)	Viscosities (mPa·S)
LP-20241	4.12	1766	17.4
LH-20243	4.01	1654	17.4
LK-20260	3.99	1712	16.9
PLS-20261	4.36	1822	19.1
LYS-20297	4.27	1824	16.0

Note: *L. paracasei* (LP-20241), *L. helveticus* (LH-20243), *L. kefiri* (LK-20260), Plant *L*. species (PLS-20261), *Lactobacillus* Yogurt Starter (LYS-20297).

**Table 2 antioxidants-13-00382-t002:** Contents of total sugars, reducing sugars, flavonoids, polyphenols, and proteins in RRT fruit fermentation broth from different strains of bacteria.

Concentration (mg/mL)	LP-20241	LH-20243	LK-20260	PLS-20261	LYS-20297
Total sugars	85.78 ± 0.005 ***	72.83 ± 0.006 ***	69.44 ± 0.002 ***	73.45 ± 0.005 ***	130.92 ± 0.009
Reducing sugars	9.21 ± 0.083 ***	7.76 ± 0.139 ***	7.51 ± 0.131 ***	6.59 ± 0.037 ***	10.50 ± 0.083
Flavonoids	1.40 ± 0.026 ***	3.14 ± 0.035 ***	1.10 ± 0.016 ***	1.40 ± 0.018 ***	1.81 ± 0.035
Polyphenols	8.75 ± 0.001 ***	11.20 ± 0.004 **	11.66 ± 0.002 ***	8.46 ± 0.002 ***	10.46 ± 0.001
Proteins	46.02 ± 0.026 *	45.00 ± 0.001 **	46.02 ± 0.012 *	44.33 ± 0.008 **	49.06 ± 0.017

Note: Values are expressed as mean ± SD. (*: *p* < 0.05, **: *p* < 0.01, ***: *p* < 0.001; other sample groups vs. LYS-20297).

**Table 3 antioxidants-13-00382-t003:** Summary of photoaging-related pathway genes.

Gene Name	Gene Description	C vs. M	M vs. R
*Hmox1*	Heme oxygenase 1	DOWN	UP
*Jun*	Jun proto-oncogene Symbol; Acc:	UP	DOWN
*Fos*	FBJ osteosarcoma oncogene	UP	DOWN
*Egfr*	Epidermal growth factor receptor	DOWN	UP
*Col4a6*	Collagen, Type IV, Alpha 6	DOWN	UP
*Col4a1*	Collagen, Type IV, Alpha 1	DOWN	UP
*Col4a2*	Collagen, Type IV, Alpha 2	DOWN	UP
*Mmp2*	Matrix metallopeptidase 2	UP	DOWN
*Tgfb2*	Transforming growth factor, Beta 2	DOWN	UP
*Bcl2*	B cell leukemia/lymphoma 2	DOWN	UP
*Mmp14*	Matrix metallopeptidase 14	UP	DOWN
*Mmp9*	Matrix metallopeptidase 9	UP	DOWN

## Data Availability

Data are contained within the article.

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
