# Peer review of "Protective Mechanism of Rosa roxburghii Tratt Fermentation Broth against Ultraviolet-A-Induced Photoaging of Human Embryonic Skin Fibroblasts"

_antioxidants, 2024, doi:10.3390/antiox13030382_

Round 1

Reviewer 1 Report

Overall, the paper is well written, well presented, and the methodologies used are pertinent.  

Two important aspects must be ammeliorated:

In the introduction, more specific information regarding the photoaging effects at skin level must be added.

Line 233 – please provide one more method to prove the antioxidant activity and compare it to a positive control.

Introduction

Lines 61 – 62: please be more specific in mentioning which type of MMPs are increased by UVA which affects specifically the collagen fibers and the other components of extracellular matrix.   

Materials and methods

Line 101 – Please mention the concentration of bacterial strains used for fermentation (CFU/ml)

Lines 104 – 114: a short description of methods used to determine total sugar, reducing sugar, flavonoids, polyphenols, and protein content should be presented, as well as the reagents used.

Lines 141-142: please add the time of UVA exposure of cell cultures.

Lines 151-152, 168-169: the experimental steps to assay cell viability should be added, even briefly; same for determination of total antioxidant capacity and lipid peroxidation level.    

Line 215: please provide de significance of abbreviations (LP, LK, LYS etc.) and make a uniformly presentation of the results (same number of decimals after 0).

Results

Line 240 – please give arguments for choosing 200 μg/ml as experimental cell concentration for cell viability.

 Line 257 – images in figure 2 are very small and difficult to observe. Please enlarge.

Line 409 – the text presented on both axis is very small, thus illegible. Please enlarge.

The conclusions of the study must be added at the end of the article.

Reviewer 2 Report

Protective Mechanism of Rosa roxburghii Tratt Fermentation Broth Against Photoaging of Fibroblasts is an interesting article on the anti-photoaging effect of fermentation broth and aims to clarify the mechanism of action of fermented RRT. The subject is a part of the current research trend concerning the fermentation process of dietary products. Undoubtedly, the weaker side of the work is the inaccurately described methodology and the lack of standards in the study of antioxidant activity. The strength is the multitude of analyses performed, which allow us to tentatively establish the mechanism of action of the broths testing of fermented products.

1. The methods of total sugar content, DNS detection, flavonoid content, polyphenol content, and DPPH activity should be better described in a way of repeating the experiments.

2. Abbreviations of strains should be added under Table 1, Figure 1, etc.

3. The SD shown in Table 1 should be less precise (limited to a maximum of 3 decimal places).

4. Figure 1 does not correlate with the values quoted in the text - all IC50s are higher than 0.1 mg/ml.

5. A reference is needed for DPPH and ABTS analysis (Figure 1). Without a referential substance, it is difficult to determine whether the activity is strong or not/ or which reference substance was used to express the results of ABTS activity Figure 1 B.

7. Why was Figure 2B prepared from a higher concentration to a lower concentration?

8. Graphs 5,6,7 are unreadable (too small).

9. Once used, the Latin name can be used later as an abbreviation, e.g. Lactobacillus paracasei and later L. paracasei.

10. The discussion should be corrected. There are no references to the literature (or there are in few). It should be improved.

11. The Conclusion should be changed. It is important to conclude the most important achievement of the research (e.g. the most active or the least).

Reviewer 3 Report

Comments and Suggestions for Authors

The manuscript entitled "Protective Mechanism of Rosa roxburghii Tratt Fermentation Broth Against Photoaging of Fibroblasts" describes the photoprotective effects on human skin fibroblasts of broths derived following fermentation of dried fruits of the above plant with several lactic acid bacteria. Although the concept of the study is interesting, there are several weaknesses of its implementation and its description in the manuscript.

The main problem is the lack of a clear description of the study protocol. Especially the time-frame of the various manipulations is not clearly stated. Based on lines 139-143, the test samples (broths) are incubated with the cells for 24 hours, then exposure to UV-A followed, and then serum-free culture medium was added, but it is not stated anywhere for how long. The authors are advised to make a simple chart or flow-diagram to help the reader understand their protocol. Moreover, the authors should explain why they have selected this specific time-frame: why they have chosen a 24-hour pre-incubation and why they removed the test samples afterwards. They also should mention how long did the exposure to UV-A lasted, in order to achieve the irradiation dose of 500 mJ/cm2.

A further problem with the study protocol is that both gene and protein expression levels seem to be studied at the same time-point (unless this was not the case, but there is not any other indication of time-points beyond the one already mentioned in lines 139-143). However, usually changes in gene expression precede the ones in protein levels (e.g. collagen accumulation is a rather slow process compared to its gene expression changes).

The whole study has been implemented using embryonic fibroblasts, however skin fibroblasts from adult donor are more appropriate to study the effects of UV irradiation (since the embryo is protected from the sunlight). The authors should verify at least some of their key experiments using human skin fibroblasts from adult donor. Otherwise, this should be mentioned in the Discussion as a limitation of their study.

Usually, signal transduction pathways, such as the Smad and the MAPK ones, are studied at the post-translational modification levels (and at shorter time-frames than gene expression). The authors should study phosphorylation of Smads, ERK, JNK, and p38 following treatment with the broths and exposure to UV-A using appropriate phospho-antibodies and Western blotting or immunofluorescence.

The authors refer to the study of proteins, such as collagen, elastin and MMPs with the term "cytokines" (line 293, "3.5. Cytokine protein content"). This is completely wrong: cytokines are molecules triggering signaling cascades, similarly to the growth factors, while collagen and elastin are extracellular matrix components, and MMPs proteolytic enzymes.

In lines 134-135, the statement "human fibroblasts in good condition during the logarithmic growth period were counted" is scientifically questionable, since 1) "good condition" should be defined more precisely and 2) obviously cells that are not in good condition are never being used for experiments.

In line 156, the abbreviation "VC" should be explained.

In the Discussion, the part included in lines 473-480 is not clearly relevant to this study. Especially refs 36 and 37 seem to be completely irrelevant, since obviously the authors have used the various strains of lactic acid bacteria under "conventional in vitro culture conditions" and not in the conditions of the natural environment.

Comments on the Quality of English Language

There are several language errors. I have selected certain examples:

In line 48, the phrase "Results 87 different flavor compounds were detected" does not make any sense

In lines 90-92, the methodology is written in the imperative, like a recipe. Usually, in Materials and Methods past tense is used, to indicate the protocols used by the authors when implementing their study.

In line 132, "confluency" should be used instead of "fusion"

In line 147, should read "500 μl of lysis buffer" instead of "500 μl of lysate", and then "When the cells were fully lysed" instead of "When the cells were fully lysated"

In lines 327-332, in various occasions the word "Samd" should be replaced by the correct "Smad"

In line 434, the phrase "select remove ROS signaling pathways" does not make any sense

In line 512, should read "to reduce" instead of "to reduces"

Round 2

Reviewer 2 Report

Thank you to the authors for clarifying questionable points. In my opinion, the amendments introduced are sufficient. I find that the article can be published in this form.

Thank you to the authors for clarifying questionable points. In my opinion, the amendments introduced are sufficient. I find that the article can be published in this form.

Author Response

It is a great honor to get your recognition of this work, and thank you for your help to improve the quality of the article.

Reviewer 3 Report

In the revised version, the authors have responded to many minor suggestions from the first reviewing round, hence the manuscript is ameliorated.

However, they have not performed any of the experiments advised (responses 2, 3, and 4) on the ground of time constraints and laboratory conditions. If the Editors provide the necessary time the authors should perform these experiments, or mention in the Discussion the reasons for not performing them and the limitations that are posed on the study.

1) Why the authors have chosen only a 24-hr time-point for all the different parameters they have studied? Usually changes in gene expression precede the ones in protein levels and activity (e.g. collagen accumulation is a rather slow process compared to its gene expression changes), so these should be studied at different time-points.

2) Regarding the choice of embryonic instead of adult skin fibroblasts, the authors have responded that "Cells derived from embryos have the greatest cell viability and proliferation potential, so we used human embryonic skin fibroblasts for our tests". I understand this technical reason, as I also understand that maybe the authors have not access to adult human skin fibroblasts. At any case, however, embryonic cells are not naturally exposed to sunlight. On the contrary, adult skin cells are more or less exposed to sunlight, so the authors should be interested for the photoprotection of this cell type. This should be included in the Discussion as a limitation of the study design.

3) As I mentioned in the first reviewing round, the authors should study phosphorylation of Smads, ERK, JNK, and p38 following treatment with the broths and exposure to UV-A using appropriate phospho-antibodies and immunofluorescence (in case they have not the necessary laboratory equipment to perform Western blotting). As they already have shown fluorescence microscopy data (e.g. Figure 2B), meaning that they have the possibility to perform immunofluorescence in their lab, the authors should perform these experiments, if the Editors are convinced to provide the necessary time.

Round 3

Reviewer 3 Report

The manuscript is now suitable for publication.

There are no further comments.